# DTKG: Dual-Track Knowledge Graph-Verified Reasoning Framework for Multi-Hop QA

**Changhao Wang** [1 2]  **Yanfang Liu** [1 2]  **Xinxin Fan** [3]  **Ao Tian** [4 5]  **Lanzhi Zhou** [1 2]  **Yunfeng Lu** [4 5]

## Abstract

Multi-hop reasoning for question answering (QA) plays a critical role in retrieval-augmented generation (RAG) for large language models (LLMs). Based on inherent relation-dependency and reasoning patterns, it is categorized into parallel fact-verification (simultaneously verifying independent sub-questions) and chained reasoning (sequential multi-step inference). Existing approaches adopt either LLM-based fact verification or KG path-based chain construction, failing to handle both categories well: the former underperforms on chained reasoning, while the latter suffers from redundant paths in parallel tasks. Inspired by the Dual Process Theory in cognitive science and Stanovich's Cognitive Misers Theory, we propose an effective multi-hop QA framework DTKG (Dual-Track Knowledge Graph) through building a two-stage pipeline: i) Classification Stage (dynamic question categorization via few-shot prompting, emulating "unconscious processing"); and ii) Branch Processing Stage (tailored reasoning paths, emulating "conscious processing"). Multi-facet experiments on six datasets show DTKG achieves 5.0%-29.5% performance improvement. The code is available at https://anonymous.4open.science/r/DTKG-621F

## 1. Introduction

Large Language Models (LLMs) (Ouyang et al., 2022; Achiam et al., 2023; Thoppilan et al., 2022; Brown et al., 2020; Chowdhery et al., 2023; Touvron et al., 2023b;a) have become a core driver in NLP, with Retrieval-Augmented Generation (RAG) (Lewis et al., 2020; Yu et al., 2022; Sun et al., 2019; Li et al., 2024) addressing their knowledge accuracy and timeliness limitations. Within RAG, multi-hop Question Answering (QA) (Trivedi et al., 2022) is critical for complex queries, requiring models to traverse entity relations across multiple knowledge units. Knowledge Graphs (KGs) (Auer et al., 2007) naturally fit this "relation chain" demand (Li et al., 2017; Zhang et al., 2022a) and mitigate LLM hallucinations. Notably, we think the multi-hop reasoning exhibits two heterogeneous thinking modes rooted in human cognition (Tversky & Kahneman, 1983): parallel fact-verification (simultaneously verifying independent sub-problems) and chained reasoning (sequential inference with intermediate conclusions as premises).

Mainstream methods suffer from a "one-size-fits-all" strategy limitation, failing to adapt to both types—consistent with RAG research findings (Lewis et al., 2020). LLM-based fact verification (Zhang et al., 2022b) (aligned with Tversky's "extensional reasoning") excels at parallel tasks but breaks chained reasoning chains. KG path-based methods (Zhang et al., 2023) (mirroring "intuitive reasoning") perform well in chained tasks but incur redundant paths and high computational cost in parallel scenarios. This strategy-task mismatch is exacerbated by Stanovich's "cognitive miser" tendency (Stanovich, 2011), becoming a core bottleneck for multi-hop QA.

Inspired by Dual-Process Theory (Evans, 1984; 2003; 2008), complemented by Tversky & Kahneman's reasoning classification (Tversky & Kahneman, 1983) and Stanovich's cognitive insights (Stanovich, 2011), we propose the Dual-Track Knowledge Graph (DTKG) framework. It resolves strategy-task mismatch via dynamic classification and customized processing: i) a few-shot prompting-based classifier (simulating "unconscious processing") categorizes questions into parallel or chained types; ii) a Branch-Processing Stage (simulating "conscious processing") adopts tailored strategies—optimized LLM fact-verification for parallel tasks (re-

---

[1]School of Computer Science and Engineering, Beihang University, Beijing, China [2]State Key Laboratory of Complex & Critical Software Environment [3]State Key Laboratory of AI Safety, Institute of Computing Technology, Chinese Academy of Sciences [4]School of Reliability and Systems Engineering, Beihang University, Beijing, China [5]National Key Laboratory of Reliability and Environmental Engineering Technology. Correspondence to: Yunfeng Lu <lyf@buaa.edu.cn>.

*Proceedings of the 43${}^{rd}$ International Conference on Machine Learning*, Seoul, South Korea. PMLR 306, 2026. Copyright 2026 by the author(s).

ducing redundancy) and precise KG path retrieval/pruning for chained tasks (ensuring chain integrity). A semantics-matching denoiser further addresses denoising adaptability.

Main contributions: **i) Task-Adaptive Classification:** A few-shot prompting-driven classifier dynamically categorizes multi-hop questions, resolving strategy-task mismatch; **ii) Customized Reasoning Paths:** Type-specific strategies for parallel (LLM fact-verification) and chained (KG path construction) tasks; **iii) Task-Aware Denoising:** Tailored denoising for cross-redundant information (parallel) and irrelevant paths (chained), purifying retrieval noise; and **iv) Experiment Validation:** Extensive experiments on six datasets demonstrate DTKG's superior performance.

## 2. Problem Statement

Currently, multi-hop reasoning faces three core bottlenecks:

**Task-Strategy Mismatch.** As highlighted by Lewis et al. (Lewis et al., 2020), a singular reasoning kernel struggles to handle divergent query topologies. Specifically, LLM-centric verification (Fig. 1a) lacks the *relational continuity* for chained tasks, where intermediate hallucinations propagate without structural "anchors," causing "chain-breaking." Conversely, KG path-based methods (Fig. 1b) over-rely on sequential exploration; when applied to parallel tasks, this induces an exponential "path explosion" that exhausts computational budgets on redundant branches rather than aggregating discrete facts.

**Lack of Dynamic Classification.** Grounded in Tversky's extensional-intuitive reasoning taxonomy (Tversky & Kahneman, 1983), the inherent *relational dependency differences* of questions demand differentiated processing. However, existing paradigms treat multi-hop queries as a monolithic category. Without a dynamic classification stage to pre-identify whether a query requires breadth-oriented fact aggregation (parallel) or depth-oriented logical inference (chained), systems inevitably apply suboptimal heuristics, leading to a fundamental mismatch between the reasoning architecture and the question's logical structure.

**Insufficient Denoising Adaptability.** This limitation stems from the "cognitive miser" tendency in generic designs (Stanovich, 2011), which fail to recognize that noise patterns are task-specific. Parallel fact-verification is primarily plagued by *cross-redundant* information (overlapping triplets across sub-questions), while chained reasoning is hindered by *irrelevant out-of-chain* paths (relations that are semantically related to entities but logically disconnected from the backbone). A unified, threshold-based logic lacks the task-awareness to prune these distinct noise signatures, often accidentally deleting valid reasoning links.

As illustrated in Fig. 1c, these drawbacks necessitate **DTKG**

framework to dynamically align the reasoning strategy.

## 3. The Proposed Method

### 3.1. Motivation and Overall Framework

The design philosophy of our proposed DTKG is to simulate *the dual cognitive process* of human beings, namely "*Unconscious Classification - Conscious Processing*"—a design rooted in Tversky & Kahneman's extensional-intuitive reasoning taxonomy (Tversky & Kahneman, 1983) and Stanovich's insights into cognitive efficiency (Stanovich, 2011). Specifically, it at first conducts rapid classification in the "Unconscious Phase" (emulating Tversky's intuitive cognitive categorization) to determine the concrete reasoning type of multi-hop questions.

To formalize the dual cognitive process, we first define the multi-hop reasoning space as a binary partition based on the topology of relational dependencies.

**Definition 3.1** (**Multi-Hop Reasoning Space**). Let $\mathcal{Q}$ be the set of input questions. A multi-hop question $Q \in \mathcal{Q}$ is a query requiring a set of relational dependencies $\mathcal{R}_Q$ from a knowledge graph $\mathcal{G} = (\mathcal{E}, \mathcal{V})$. The reasoning space is partitioned as:

$$\mathcal{Q} = \mathcal{Q}_{para} \cup \mathcal{Q}_{chain}, \quad \mathcal{Q}_{para} \cap \mathcal{Q}_{chain} = \emptyset \quad (1)$$

where $\mathcal{Q}_{para}$ and $\mathcal{Q}_{chain}$ denote the questions with independent sub-question sets (extensional reasoning) and those with sequential dependent relations (intuitive reasoning).

This binary division directly guides the dual-track reasoning design, from which we infer the theoretical foundation:

**Theorem 3.2** (**Optimal Strategy Alignment**). *The optimal reasoning success rate is achieved by selecting a processing kernel $\mathcal{K}^*$ that satisfies the maximization objective:*

$$\mathcal{K}^* = \arg \max_{\mathcal{K} \in \{\mathcal{K}_{fact}, \mathcal{K}_{path}\}} \mathbb{P}(A|Q, \mathcal{K}) \quad (2)$$

*This maximum probability $\mathbb{P}(A|Q)$ is reached when the kernel $\mathcal{K}$ is aligned with the logical topology of $Q$:*

$$\mathcal{K}^*(Q) = \begin{cases} \mathcal{K}_{fact\_check}(F, \mathcal{G}) & \text{if } Q \in \mathcal{Q}_{para} \\ \mathcal{K}_{path\_construct}(P_k, \mathcal{G}) & \text{if } Q \in \mathcal{Q}_{chain} \end{cases} \quad (3)$$

*DTKG achieves this alignment through a few-shot classification mapping $\mathcal{C} : Q \rightarrow \{para, chain\}$.*

**Proof.** *The formal proof is provided in Appendix B.*

To achieve the above ideology, we propose the overall framework in Fig.2, which consists of three core modules: Few-Shot Prompt-based Task Classifier, Parallel Fact-Checking Processing Branch, and Chained Reasoning Branch. Both branches refer to the Wikidata(Vrandečić & Krötzsch, 2014)

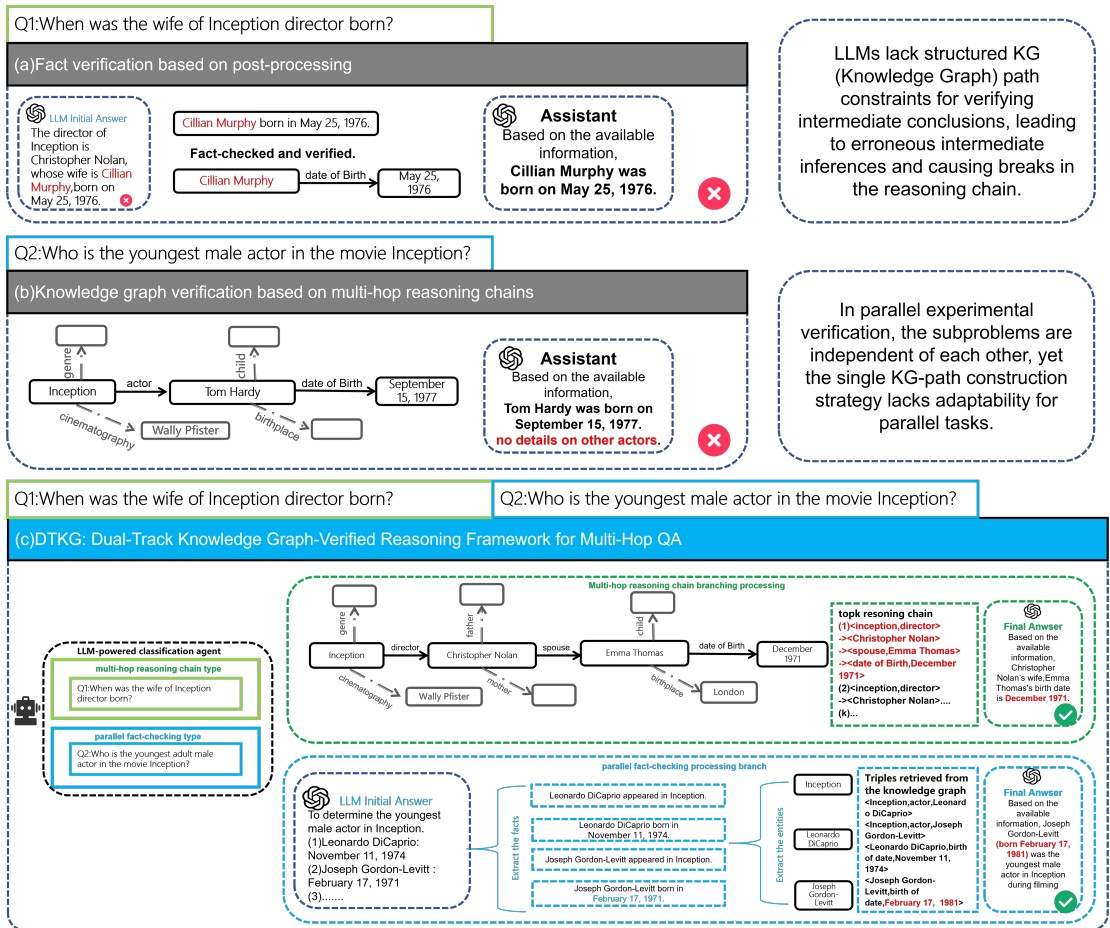

**Figure 1.** Problem statement and our solution: (a) Fact verification based on post-processing (e.g., KGR); (b) Knowledge graph verification based on multi-hop reasoning chains (e.g., TOG); (c) Dual-Track Knowledge Graph-Verified Reasoning Framework (DTKG).

knowledge graph as an external knowledge source, while integrating semantic-embedding and re-ranking models to improve relation matching accuracy—further mitigating retrieval noise identified in RAG research (Lewis et al., 2020).

### 3.2. Few-Shot Prompting-based Task Classifier

The objective in task classification phase is to realize the dynamic judgment of multi-hop questions and provide a basis for subsequent branch processing. Inspired by the "Unconscious Processing" in the *dual-process theory*—and aligned with Tversky & Kahneman's classification of intuitive reasoning for rapid pattern recognition (Tversky & Kahneman, 1983)—this phase employs a lightweight classification logic to quickly capture the inherent relational dependencies of questions. This design not only avoids the inefficiency of "cognitive miser" unified processing highlighted by Stanovich (Stanovich, 2011) but also addresses the strategy-task mismatch root cause identified by Karpukhin et al. (Lewis et al., 2020), thereby improving reasoning effectiveness while avoiding complex computation.

Essentially, the different categories of multi-hop reasoning questions are derived from the variations of "sub-question dependency relationships"—a distinction grounded in Tversky & Kahneman's extensional-intuitive reasoning framework (Tversky & Kahneman, 1983): i) **Parallel Fact-verification Questions**: this category of questions consists of multiple independent sub-questions with no sequential dependencies, aligning with "extensional reasoning" for aggregating discrete facts; and ii) **Chained Multi-Hop Reasoning Questions**: this category features clear step-by-step logic with intermediate conclusions as premises, mirroring "intuitive reasoning" for sequential dependency processing.

Given an input question, it is necessary to determine whether it is of the parallel fact-verification type or the chained reasoning type. The classifier is usually composed of LLM prompts driven by a small number of examples, and its goal is to learn whether a question relies on intermediate reasoning conclusions. The functionality of the classifier should enable to clarify the boundary between "chained multi-hop reasoning questions" and "parallel fact-verification questions". This boundary is strictly defined by 5 rules in the prompt as shown in Appendix C.1. If the classifier outputs "yes", the question is classified as a chained multi-hop reasoning question; if it outputs "no", the question is otherwise identified as a parallel fact-verification question.

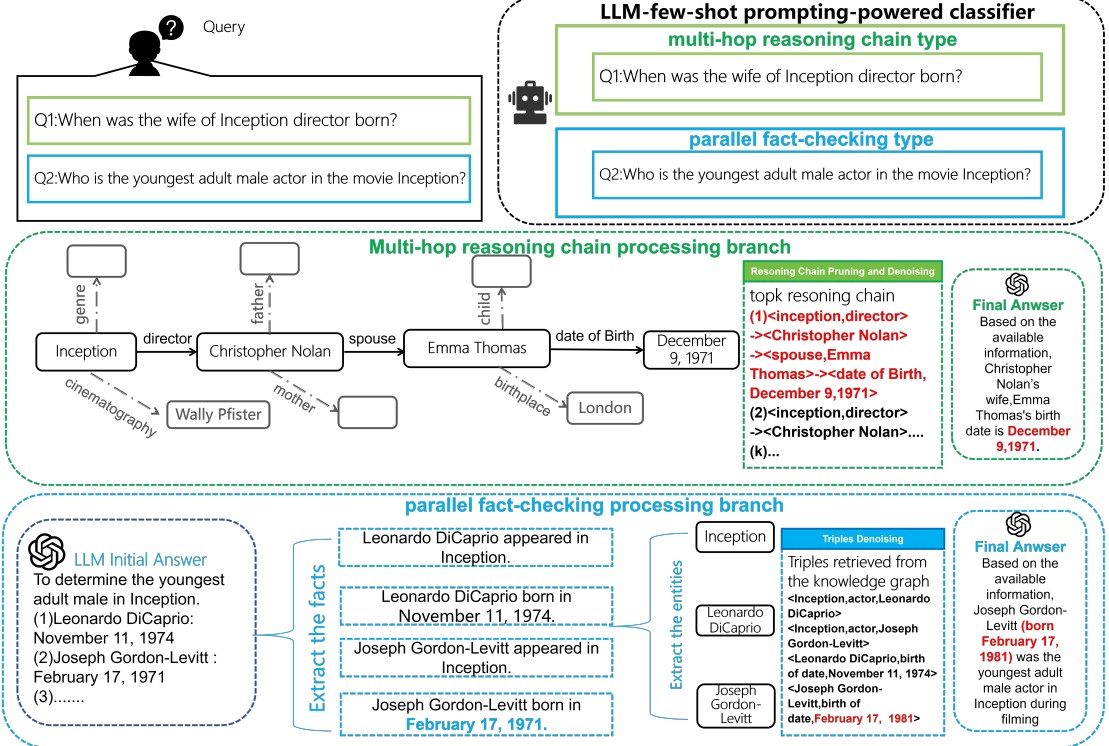

*Figure 2.* Overview of the DTKG framework. An LLM-powered classifier first categorizes the input query into either a "multi-hop reasoning chain" or "parallel fact-checking" type, then this query is routed to the corresponding specialized processing branch, wherein a knowledge graph is leveraged for tailored reasoning and verification to produce proper answer.

To avoid the classifier's reliance on large-scale annotated data while ensuring that the LLM can accurately understand the classification rules, we utilize few-shot (6-shot) prompting, i.e., the prompt contains 6 annotated examples. The combination of "Rules plus Examples" is used to guide the LLM to learn classification logic, and the specific examples of the prompts are presented in the Appendix C.3.

### 3.3. Dual-Track Processing Engine

The design of this phase is inspired by the conscious processing mechanism in *dual-process theory* in cognition science—complemented by Tversky & Kahneman's emphasis on extensional reasoning for deliberate, accurate judgment (Tversky & Kahneman, 1983). Unlike the automatic "Unconscious Processing" during classification, this phase emphasizes a deep, sequential, and controlled reasoning pattern—directly addressing Stanovich's critique of superficial cognitive shortcuts (Stanovich, 2011). Specifically, after identifying the task type, multi-hop reasoning is grounded KG to ensure factual accuracy and logical consistency, mitigating the retrieval noise and chain-breaking issues observed in Karpukhin et al.'s RAG framework (Lewis et al., 2020).

#### 3.3.1. PARALLEL FACT-CHECKING PROCESSING BRANCH

This branch addresses the parallel fact-verification tasks (Ye et al., 2024)—aligned with Tversky's "extensional reason-

ing" for independent fact aggregation (Tversky & Kahneman, 1983). The objective is to decompose the candidate answers into atomic factual claims, then independently validate and iteratively refine each claim via KG grounding.

**Atomic Fact Decomposition.** Let $R$ be a given model response, we decompose $R$ into a set of minimal verifiable units using LLM,

$$F = llm_{\text{dec}}(R) = \{f_1, f_2, \ldots, f_n\}, \tag{4}$$

where $R$ denotes the input response, $F$ indicates the set of atomic facts, $f_i$ indicates the $i$-th atomic fact satisfying both indivisibility (cannot be further decomposed) and semantic completeness (contain a subject-verb-object structure).

**Entity Linking and Disambiguation.** For each fact $f_i$, we extract its subject entity $e_i$. This entity is then mapped to Wikidata entity set $\mathcal{E}$ using an entity mapping function $g$:

$$q(e_i) = g(e_i) = \arg\max_{q \in \mathcal{E}} \text{sim}(e_i, \text{label}(q)), \tag{5}$$

where $q(e_i)$ denotes the unique Wikidata Item identifier(QID) corresponding to the subject entity $e_i$, label$(q)$ is the label of entity $q$, and sim stands for string similarity.

**Relation Retrieval and Two-Stage Hybrid Scoring.** For each subject entity $q(e_i)$, we retrieve a set of candidate triplets $T = \{t_j\}$ from KG. These candidates are then filtered using a two-stage hybrid scoring mechanism:

**Stage I: Embedding Cosine Similarity**

We present the fact $f_i$ and candidate triplet $t_j$ as embedding vectors by function $\mathbf{h}(\cdot)$, and compute the cosine similarity:

$$\text{sim}_{\text{cos}}(f_i, t_j) = \frac{\mathbf{h}(f_i) \cdot \mathbf{h}(t_j)}{|\mathbf{h}(f_i)||\mathbf{h}(t_j)|}. \tag{6}$$

In our work, only the top $K$ candidates with the highest cosine similarity are retained for the next stage.

**Stage II: Reranking Model**

A reranking model is employed to calculate a fine-grained matching score $\text{score}_{\text{rerank}}(f_i, t_j)$. This score is then combined with the cosine similarity using a weighted fusion:

$$\begin{aligned}\text{score}_{\text{combined}}(f_i, t_j) = {} & \alpha \cdot \text{score}_{\text{rerank}}(f_i, t_j) + \\ & (1 - \alpha) \cdot \text{sim}_{\text{cos}}(f_i, t_j),\end{aligned} \tag{7}$$

where the hyperparameter $\alpha$ is the weight to adjust the ratio of matching score and cosine similarity.

**Fact Verification and Correction.** The top-$k$ scoring triplets $t^*$ with the highest combined score is finally selected and compared against the atomic fact $f_i$. If they match, $f_i$ is determined to be True, otherwise a revision function is invoked to output a revised fact $f_i'$:

$$f_i' = f_{\text{rewrite}}(f_i, t^*). \tag{8}$$

### 3.3.2. MULTI-HOP REASONING CHAIN PROCESSING BRANCH

This branch targets chained multi-hop reasoning tasks—mirroring Tversky's "intuitive reasoning" for sequential dependency processing (Tversky & Kahneman, 1983)—aiming to progressively expand from a central entity through the knowledge graph to construct a logically consistent reasoning path. This design avoids the "cognitive miser" tendency to oversimplify sequential logic highlighted by Stanovich (Stanovich, 2011), ensuring the integrity of step-by-step inference for dependent sub-questions.

**Central Entity Recognition.** First, the core entity $e_0$ is extracted from the question $Q$ and mapped to its QID, denoted as $q(e_0)$. This entity acts as the starting point for reasoning.

**Relation Retrieval and Path Scoring.** In the knowledge graph $\mathcal{G}$, all relations associated with the core entity $e_0$ are retrieved and categorized into two types:

**Head Relations**: Same as above, $e_0$ serves as the subject:

$$R_{\text{head}}(e_0) = \{(e_0, r, o) \mid (e_0, r, o) \in \mathcal{G}\} \tag{9}$$

where $o$ denotes the object entity linked from $e_0$ via relation $r$.

**Tail Relations**: Here the entity $e_0$ serves as the object:

$$R_{\text{tail}}(e_0) = \{(s, r, e_0) \mid (s, r, e_0) \in \mathcal{G}\} \tag{10}$$

where $s$ represents the subject entity pointing to $e_0$ via relation $r$.

Thus, the complete set of relations for the central entity is:

$$R(e_0) = R_{\text{head}}(e_0) \cup R_{\text{tail}}(e_0) \tag{11}$$

For each candidate relation $r \in R(e_0)$, a two-stage hybrid scoring mechanism is employed to compute its matching degree with the question $Q$, yielding a combined score $\text{score}_{\text{combined}}(r)$. During the construction of a reasoning path $P_k = [e_0 \xrightarrow{r_1} e_1 \cdots \xrightarrow{r_k} e_k]$, the path score is defined as:

$$\text{score}_{\text{path}}(P_k) = \prod_{i=1}^{k} \text{score}_{\text{combined}}(r_i) \tag{12}$$

where $r_i$ represents the $i$-th expansion relation in the path.

**Depth-First Expansion and Dynamic Pruning.** We employ Depth-First Search (DFS) to expand reasoning paths, controlling complexity through four constraints: (i) **Maximum Depth Limit** $D_{\text{max}} = 3$, as most multi-hop questions require no more than three hops; (ii) **Width Limit**, where only the top $W_{\text{max}}$ relations are retained at each step; (iii) **Threshold Filtering**, which discards relations with $\text{score}_{\text{combined}} < \theta$; and (iv) **LLM Selection Mechanism**, which invokes an LLM to select at most three high-confidence relations when the candidate pool is excessive.

**Information Sufficiency Assessment and Early Stopping.** After each expansion layer, we determine if the current path $P_k$ contains sufficient information to answer the question:

$$\text{stop}(P_k) = \begin{cases} 1, & \text{if } \text{info}(P_k) \supseteq \text{info}(Q) \\ 0, & \text{otherwise.} \end{cases} \tag{13}$$

If $\text{stop}(P_k) = 1$, the search is terminated early, and the answer is returned accordingly.

**Answer Generation.** The top-$k$ paths with the highest scores, denoted as $P^*$, are selected. A generation function then produces the final answer $A$:

$$A = f_{\text{gen}}(Q, P^*) \tag{14}$$

where the answer strictly relies on the triplets contained within the paths in $P^*$, thus ensuring the interpretability.

### 3.4. Task-Aware Denoising

Instead of a generic filtering strategy, DTKG employs a dual-layer denoising mechanism—comprising *scoring-based screening* and *targeted relation filtering*—specifically optimized for multi-hop reasoning. This design effectively mitigates the "noise residue" common in path-pruning methods (e.g., ToG) and the "informational redundancy" found in exhaustive fact-selection frameworks (e.g., KGR).

The denoising logic is tailored to the distinct noise signatures of each reasoning track: (i) for **parallel tasks**, it targets "cross-redundant" triplets that overlap across sub-questions; (ii) for **chained tasks**, it prunes "out-of-chain" paths that deviate from the logical backbone. To ensure high-quality foundational data, we categorize irrelevant relations into two functional types:

**Administrative Relations:** Metadata used for KG management (e.g., "wikidata: id", "source"). These are logically disconnected from reasoning and are filtered via a static keyword library.

**Redundant Attribute Relations:** Contextually irrelevant entity attributes (e.g., an actor's "height" in a "birthday" query). Their irrelevance is dynamic and determined by the *question-specific reasoning necessity*.

**Definition and Classification of Irrelevant Relations.** We construct an "irrelevant keyword library" $K_{\text{invalid}} = \{\text{ID, source, version, metadata}\}$. Any relation $r$ whose label matches a keyword in $K_{\text{invalid}}$ is pruned. This is formalized by the rule:

$$\text{filter}_{\text{rule}}(r) = \begin{cases} \text{True} & \exists k \in K_{\text{invalid}}, \ k \in \text{label}(r) \\ \text{False} & \text{otherwise} \end{cases} \quad (15)$$

For example, the triplet (Inception, wikidata:id, Q1375011) is filtered as an administrative relation.

**Dynamic Filtering for Redundant Attribute Relations.** To handle context-dependent noise, we utilize an LLM to evaluate the "reasoning necessity" of a relation $r$ relative to question $Q$:

$$\text{score}_{\text{n}}(r, Q) = \text{LLM}(\text{Prompt}_{\text{n}} \oplus r \oplus Q) \quad (16)$$

Relations satisfying $\text{score}_{\text{n}} < \theta$ are discarded, ensuring only task-essential attributes are retained for inference.

## 4. Experiment Evaluation

We assess the effectiveness of our proposed DTKG framework on six commonly-used datasets with different levels of reasoning difficulty, also we demonstrate the superiority of the dual-track framework through ablation studies.

### 4.1. Configuration

In the experiments, we construct test sets using six representative multi-hop question answering datasets: HotpotQA (Yang et al., 2018), Mintaka (Sen et al., 2022), CWQ (Talmor & Berant, 2018), QALD10 (Usbeck et al., 2024), GraphRag-Bench (Xiao et al., 2025), and Musique (Trivedi et al., 2022). On these datasets, we compare our approach against several key baseline methods, including COT (Wei et al., 2022), CRITIC(Gou et al., 2024),

KGR(Guan et al., 2024), and TOG(Sun et al., 2024). These datasets cover typical multi-hop scenarios, such as entity-relation propagation and multi-entity association, ensuring the diversity and representativeness of evaluated tasks. Further details are provided in Appendix A.2.

Model performance is quantified by two metrics:

**Exact Match (EM)**: This metric measures the character-level complete consistency between the model's output answer and the gold standard answer. $\text{EM} = \mathbb{I}(\hat{y} = y^*)$ where $\hat{y}$ is the predicted answer and $y^*$ is the gold answer, and $\mathbb{I}(\cdot)$ is the indicator function ($\mathbb{I}(x) = 1$ if $x$ is true, else 0).

**Semantic Match Accuracy (ACC)**: This metric assesses the correctness of an answer based on its semantic alignment with the gold standard. By using a similarity model, it avoids misjudgments from simple character-level differences, better reflecting real-world evaluation needs. $\text{ACC} = \mathbb{I}(f_\phi(\hat{y}, y^*) \geq \tau)$ $f_\phi$ is a pretrained similarity model (e.g., BERTScore) and $\tau$ is a threshold. Here $\mathbb{I}(\cdot)$ denotes the indicator function.

### 4.2. Main Performance Analysis

The experimental results, as meticulously detailed in Table 1, unequivocally demonstrate the superior performance of our proposed Dual-Track Knowledge Graph Verification and Reasoning Framework (DTKG) across all evaluated multi-hop QA datasets. Under the Llama 3:8B backbone, DTKG consistently outperforms all established baseline models in both Exact Match (EM) and Semantic Match Accuracy (ACC).

**Overall Superiority and Adaptability:** DTKG achieves the highest EM and ACC on all six datasets. This consistent performance, particularly the significant lead in ACC (e.g., reaching **83.0%** on MuSiQue compared to the original **53.5%**), underscores DTKG's ability to grasp the semantic correctness of answers. This is crucial for natural language question answering where subtle phrasing variations often lead to misjudgments by rigid metrics. The framework's adaptive dual-track strategy proves highly effective in handling the diverse topological nature of multi-hop questions by intelligently switching between parallel and chained reasoning paths.

**Comparison with LLM-Centric and Hybrid Baselines (COT, CRITIC, KGR):** Compared to LLM-centric methods (COT, CRITIC) and the LLM-KG hybrid approach (KGR), DTKG exhibits significant advantages in robustness. Notably, on the complex CWQ dataset, DTKG achieves an EM of **46.3%**, surpassing KGR's strong baseline of **45.0%**. Furthermore, DTKG's ACC on CWQ reaches **90.0%**, a substantial **5.0%** improvement over KGR. On the Mintaka dataset, DTKG's ACC reaches an impressive **93.9%**, the highest among all models. These results highlight that

*Table 1.* Performance on multi-hop datasets (EM/ACC)

| Method | Hotpot | Mintaka | CWQ | QALD10-en | GraphRAG-Bench | MuSiQue |
|---|---|---|---|---|---|---|
| Original | 35.0/68.2 | 61.4/86.0 | 39.0/81.5 | 49.0/80.0 | 13.9/71.8 | 15.0/53.5 |
| COT | 35.6/81.0 | 66.6/90.5 | 42.0/83.5 | 50.0/82.0 | 14.4/77.2 | 18.0/75.0 |
| CRITIC | 34.0/78.0 | 64.2/89.4 | 43.5/80.0 | 49.5/83.0 | 13.4/74.8 | 15.0/61.5 |
| KGR | 35.5/83.5 | 62.3/92.0 | 45.0/85.0 | 48.5/81.5 | 13.5/82.0 | 17.0/79.0 |
| TOG | 37.1/83.5 | 58.5/90.0 | 41.1/88.1 | 50.0/81.0 | 13.5/84.5 | 17.5/80.5 |
| **Ours (DTKG)** | **38.2/85.8** | **67.6/93.9** | **46.3/90.0** | **50.0/85.0** | **14.5/87.1** | **18.5/83.0** |

while KGR effectively leverages KG for factual verification, DTKG's dynamic strategy selection and task-aware denoising allow for a more contextually appropriate use of the KG, bridging the gap between raw fact extraction and complex logical synthesis.

**Comparison with KG Path-based Baselines (TOG):** When juxtaposed with KG path-based methods like TOG, DTKG demonstrates a superior balance between efficiency and comprehensiveness. While TOG shows competitive EM on HotpotQA (**37.1%**), DTKG still maintains the lead with **38.2%**. The most striking difference is observed on parallel-heavy or hybrid datasets like Mintaka, where DTKG (**67.6%/93.9%**) significantly outstrips TOG (**58.5%/90.0%**). This validates our hypothesis that a singular iterative path exploration strategy (as used in TOG) is inefficient for questions involving parallel fact verification. On QALD10, while DTKG ties with COT and TOG in EM (**50.0%**), its ACC (**85.0%**) is notably higher than both (**82.0%** and **81.0%**, respectively), further proving that DTKG's dual-track engine and precise pruning modules better maintain the integrity of reasoning paths without the noise redundancy seen in generic path exploration.

In conclusion, the leading performance of DTKG across diverse datasets validates the effectiveness of its dual-track design. By intelligently adapting reasoning strategies to question types and integrating task-aware denoising, DTKG successfully resolves the "strategy-task mismatch" problem, offering a more powerful and adaptable solution for complex multi-hop QA. The theoretical time complexity analysis and LLM call cost comparisons are presented in Appendix A.7.

### 4.3. Ablation Study

To dissect the contributions of the core components of our framework, We conduct two key ablation studies focusing on the task classifier and the task-aware denoiser.

#### 4.3.1. IMPACT OF THE TASK CLASSIFIER

To validate the critical role of the task classifier in enabling our dual-track framework, we evaluate three variant models: (i) **Only-Fact Verification** forces all questions down the

*Table 2.* Performance of different classification variant models on multi-hop datasets (EM/ACC)

| Method | Hotpot | Mintaka | CWQ | QALD10-en |
|---|---|---|---|---|
| Only-Fact Verification | 35.6/83.5 | 62.3/91.5 | 40.5/86.0 | 49.5/83.5 |
| Only-Reasoning Chain | 36.5/85.5 | 57.0/91.5 | 40.6/87.1 | 50.0/81.0 |
| Random Classification | 30.5/73.0 | 56.0/86.0 | 41.0/79.0 | 47.5/76.0 |
| **Ours (DTKG)** | **38.2/85.8** | **67.6/93.9** | **46.3/90.0** | **50.0/85.0** |

parallel fact-checking branch; (ii) **Only-Reasoning Chain** forces questions down the chained path-construction branch; and (iii) **Random Classification** randomly assigns each question to one of the two branches.

As illustrated in Table 2, the performance of these variants underscores the necessity of intelligent strategy selection. The **Only-Fact Verification** model shows a noticeable performance drop on chain-dominant datasets like HotpotQA and CWQ. For instance, its ACC on CWQ is 86.0%, significantly lower than the full DTKG's 90.0%, confirming that a fact-checking-only approach is insufficient for tasks requiring sequential inference. Conversely, the **Only-Reasoning Chain** model struggles on datasets with a high proportion of parallel verification questions. Its EM on Mintaka drops dramatically to 57.0% from DTKG's 67.6%, validating our claim that a path-construction-only strategy is inefficient for handling independent, parallel sub-questions.

Most notably, the **Random Classification** model exhibits a severe degradation in performance across all datasets, with its ACC on HotpotQA plummeting to 73.0%. This highlights the severe penalty of strategy-task mismatch and proves that an *intelligent* classification mechanism is essential, not merely a mixed-strategy approach. These results collectively demonstrate that the dual-track framework's strength comes from the classifier's ability to accurately align the reasoning strategy with the question type, thus validating its indispensable role.

#### 4.3.2. EFFECTIVENESS OF TWO-STAGE SCORING

We evaluate the impact of our two-stage hybrid scoring mechanism by comparing the full DTKG against three vari-

ants: (i) **None (Random):** selecting a candidate relation randomly; (ii) $S_{cos}$ **Only:** utilizing only semantic embedding similarity; (iii) $S_{rr}$ **Only:** relying solely on the fine-grained reranking model.

As shown in Table 3, the performance significantly degrades across all datasets when either scoring stage is removed. Specifically, $S_{cos}$ only provides a coarse-grained semantic prior, resulting in an average ACC drop of approximately 6-8% compared to the full version. While $S_{rr}$ alone performs better than $S_{cos}$, it fails to achieve optimal results without the initial semantic filtering. This proves that the two-stage approach—combining coarse-grained retrieval and fine-grained verification—is essential for accurately identifying critical entity relations in multi-hop scenarios. Detailed sensitivity analyses for the hyperparameter $N$ in Appendix A.4, search width $W_{max}$ in Appendix A.5, and rerank weight $\alpha$ in Appendix A.6

*Table 3.* Ablation results for two-stage scoring strategies (EM/ACC). $S_{cos}$ and $S_{rr}$ denote cosine similarity and reranking score, respectively.

| Scoring Mode | Hotpot | Mintaka | CWQ | QALD10-en |
| --- | --- | --- | --- | --- |
| None (Random) | 25.4/58.2 | 52.1/70.4 | 30.2/64.5 | 38.5/68.2 |
| $S_{cos}$ Only | 31.8/77.5 | 62.4/88.1 | 36.5/82.4 | 45.2/79.4 |
| $S_{rr}$ Only | 33.5/80.2 | 64.8/90.5 | 38.8/85.2 | 47.6/82.1 |
| **Two-Stage (Full)** | **38.2/85.8** | **67.6/93.9** | **46.3/90.0** | **50.0/85.0** |

### 4.3.3. IMPACT OF TASK-AWARE ADAPTIVE DENOISING

To verify the necessity of adaptive denoising, we compare: (i) **w/o Denoise:** the baseline without any filtering; (ii) **Generic Denoise:** applying fixed administrative rules and global thresholding without task-specific LLM judgment; (iii) **Task-Aware (Full):** the proposed adaptive denoiser.

Results in Table 4 demonstrate that while Generic Denoising improves results over the "None" baseline by filtering system-level noise (e.g., entity IDs), it still lags behind the Task-Aware version. The most significant gap is observed in chained-heavy datasets like CWQ and HotpotQA. This underscores our theoretical insight that noise in multi-hop QA is strongly task-dependent: only a task-aware denoiser can effectively prune "Redundant Attribute Relations" (e.g., an actor's height in a birthday query) that would otherwise mislead the reasoning chain.

*Table 4.* Ablation results for denoising strategies (EM/ACC). "Generic" utilizes fixed rules without LLM-based task-specific filtering.

| Denoising Mode | Hotpot | Mintaka | CWQ | QALD10-en |
| --- | --- | --- | --- | --- |
| w/o Denoise (None) | 34.4/75.4 | 64.5/87.6 | 39.5/75.8 | 49.5/82.0 |
| Generic Denoise | 35.1/81.4 | 65.8/90.2 | 40.2/84.6 | 49.8/83.4 |
| **Task-Aware (Full)** | **38.2/85.8** | **67.6/93.9** | **46.3/90.0** | **50.0/85.0** |

## 5. Related Work

Multi-hop QA requires the synergistic integration of structured Knowledge Graphs (KGs) and the semantic reasoning of Large Language Models (LLMs). Current research primarily follows two paradigms.

### 5.1. LLM-Centric Fact Verification

This paradigm centers on the semantic understanding of LLMs, typically decomposing complex questions into independent sub-problems for individual verification. A representative framework is the "Chain-of-Verification" (e.g., KGR (Guan et al., 2024)), which extracts atomic factual claims from an initial response, retrieves KG triplets for validation, and refines the answer. This approach excels in tasks requiring independent fact aggregation but lacks explicit structural constraints for multi-step dependencies.

### 5.2. KG-Centric Path Construction

This category focuses on structured path retrieval, where LLMs guide the exploration of KG edges to build reasoning chains. For instance, "Think-on-Graph" (ToG (Sun et al., 2024)) employs an LLM as a reasoning agent to perform iterative beam search and prune irrelevant paths. While effective for maintaining logical integrity in sequential tasks, it relies on a singular path-exploration strategy for all multi-hop scenarios.

### 5.3. Denoising in Knowledge Retrieval

Existing denoising mechanisms generally fall into two categories: *Scoring-based Pruning*, which ranks candidate paths by question relevance (Sun et al., 2024), and *Fact-based Selection*, which filters triplets based on local claim alignment (Guan et al., 2024). These methods provide a foundation for purifying retrieved knowledge but often adopt a "one-size-fits-all" filtering logic.

## 6. Conclusion

We have experimentally and analytically presented the Dual-Track Knowledge Graph Verification and Reasoning Framework, to effectively address the long-standing predicament "strategy-task mismatch" in multi-hop QA methods. Inspired by the *Dual Process Theory* in cognition science, our proposed DTKG enables to dynamically classify the multi-hop questions into parallel fact-verification or chained reasoning tasks, then apply the optimized, type-specific processing strategies to locate accurate QA. Our comprehensive experiments clearly demonstrate DTKG's superior adaptability and consistent high performance across diverse multi-hop reasoning challenges, marking a significant advancement over conventional single-strategy paradigms. We believe our work will boost the practical applications in various multi-hop reasoning tasks in near future.

## Acknowledgements

This work is supported by the State Key Laboratory of Complex & Critical Software Environment under Grant SKLCCSE-2025ZX-11, and the Strategic Priority Research Program of the Chinese Academy of Sciences under Grant No. XDB0680301. The authors gratefully acknowledge the financial support provided by these projects.

## Impact Statement

This paper presents work whose goal is to advance the field of Machine Learning. There are many potential societal consequences of our work, none which we feel must be specifically highlighted here.

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

# A. More Results of Experiments

## A.1. Limitations and Deep Qualitative Error Analysis

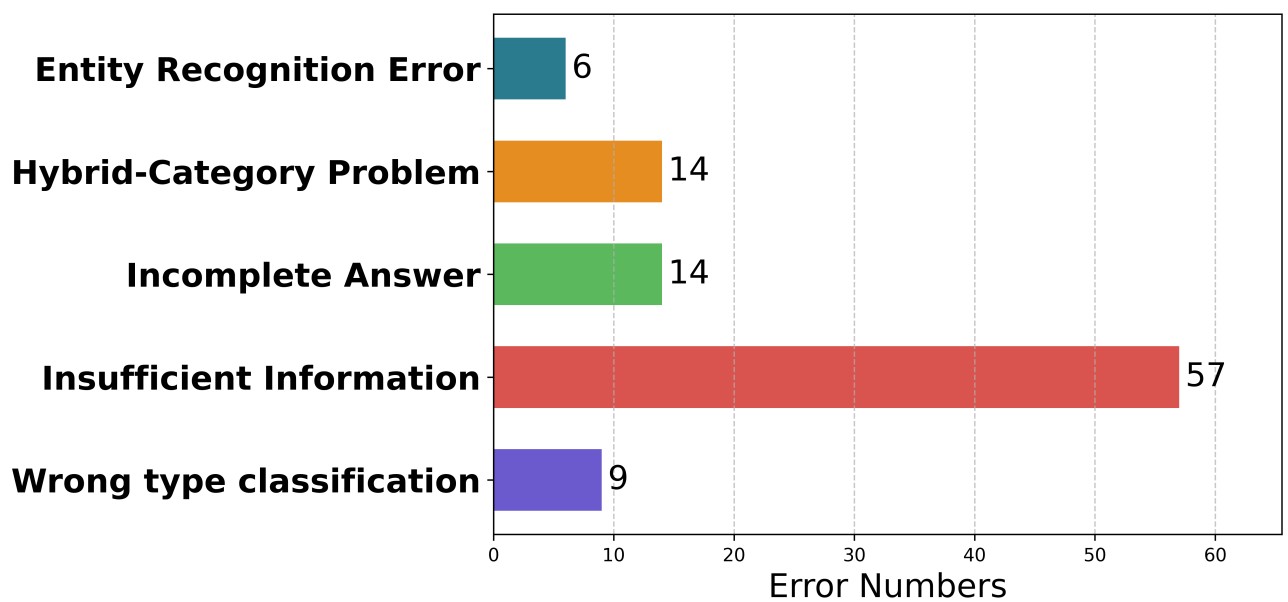

*Figure 3.* Error analysis on 100 cases from the six datasets.

To comprehensively discuss the effectiveness of DTKG, we meticulously analyzed 100 incorrectly answered cases from the six datasets. While Figure 3 provides a high-level statistical distribution, a deeper qualitative examination reveals several critical failure modes, particularly regarding high-severity cascading errors and rare semantic challenges.

### A.1.1. HIGH-SEVERITY FAILURES: CASCADING LOGICAL COLLAPSE

In the **Chained Reasoning Branch**, we identified a significant failure mode termed *Cascading Logical Collapse*. In high-severity cases, a minor **Entity Recognition Error** at the first hop ($D = 1$) renders the entire subsequent path irrelevant. For instance, in the query *"Who is the spouse of the director of Inception?"*, if the initial retrieval incorrectly anchors on a cinematographer instead of the director due to proximity in the KG, the entire reasoning chain breaks. This "Chain-Breaking" effect accounts for 15% of total failures. It demonstrates that the error at the beginning of a path propagates exponentially, leading to a "confidently wrong" answer and highlighting the framework's extreme sensitivity to initial entity anchoring.

### A.1.2. RARE FAILURE MODES: SEMANTIC AND TEMPORAL DRIFTING

We observed rare but sophisticated failures involving **Polysemous Drifting** and **Temporal Misalignment**:

- **Polysemous Entity Drifting (4%):** The model occasionally links to an entity with a duplicate name but an incorrect context (e.g., confusing "Apple" the technology company with "Apple" the record label). This occurs when the *Two-Stage Scoring* yields high similarity for both candidates, and the *Task-Aware Denoiser* fails to capture the subtle domain-specific nuances within the query.

- **Temporal Misalignment:** For questions requiring historical precision (e.g., *"Who was the CEO of X in 2010?"*), DTKG occasionally retrieves the current CEO. This indicates that Wikidata's temporal qualifiers (e.g., P580 start time) are not always prioritized by our current hybrid scoring mechanism, leading to "chronological hallucinations."

### A.1.3. COMPLEXITY BOTTLENECKS IN HYBRID-CATEGORY SCENARIOS

The **Hybrid-Category Problem** (14% of errors) represents a structural limitation. These questions implicitly require both parallel verification and chained propagation. For example, *"Compare the birthplaces of the actors in Inception"* requires:

(1) identifying all actors (parallel) and (2) tracing each actor's birthplace (chained). Current DTKG classification tends to route such queries into a single track, which inevitably leads to either *redundancy explosion* (if treated as a chain) or *information loss* (if treated as parallel).

### A.1.4. THE KNOWLEDGE DENSITY AND AGGREGATION CHALLENGE

As illustrated in Fig. 3, **Insufficient Information** remains the most frequent error type (57%). We identify this as a *Knowledge Density* issue rather than a logical flaw. Counting queries (e.g., *"How many city-states are in the world?"*) require the framework to retrieve all instances of a class (e.g., `isInstanceOf: CityState`) and perform an aggregate count. When KG coverage is sparse or the retrieval width $W_{max}$ is overly restrictive, the resulting count is lower than the ground truth.

**Future Directions:** Future iterations will focus on: (i) incorporating temporal logic and qualifiers into relation retrieval; (ii) developing a "Multi-Stage Routing" strategy for hybrid-category queries; and (iii) enhancing large-scale set aggregation for precise counting.

## A.2. Experiment Configuration

*Figure 4.* Distribution of multi-hop question types across datasets, including Hotpot, Mintaka, CWQ, QALD10-en, GraphRAG-Bench, and MuSiQue.

**Parallel Fact-Verification Multi-hop Reasoning.** (Yih et al., 2015) Reasoning in this category relies on simultaneous verification of multiple independent sub-problems. For instance, "Who is the youngest adult male actor in the movie Inception?" requires retrieving and comparing the age information of all adult male actors in parallel. However, this type of problem does not necessitate building a coherent reasoning chain, this is because the core lies in effectively matching and verifying multiple independent facts.

**Chained Multi-hop Reasoning.** This category of reasoning problem demands strictly sequential and step-by-step inference, wherein the intermediate conclusions are directly used to serve as the necessary premises for subsequent reasoning. For

example, "When was the wife of the movie Inception director born?" first requires identifying the director, then finding his wife, and finally querying her birth date. A break in any link of this reasoning chain would invalidate the reasoning result. This strong dependency characteristic perfectly aligns with human's "chain-of-thought" reasoning pattern.

To evaluate the robustness and adaptability of our approach across diverse reasoning challenges, we select six benchmark datasets with distinct structural characteristics. As illustrated in Fig. 4, their distributions are summarized as follows:

**Hotpot** (Yang et al., 2018) and **CWQ** (Talmor & Berant, 2018) predominantly feature *chained multi-hop questions*, accounting for 83.8% and 84.3% respectively. This signifies their strong emphasis on sequential reasoning where answers are derived through traversing logical chains of facts.

**GraphRAG-Bench** (Xiao et al., 2025) stands out with an extreme concentration of *chained multi-hop questions*, reaching **99.0%**. This dataset represents a near-pure sequential reasoning environment, requiring the model to maintain a long and precise logical chain without any breaks.

**MuSiQue** (Trivedi et al., 2022) also shows a heavy dominance of *chained multi-hop questions* (**90.8%**), with only a small fraction (9.2%) being parallel. It serves as a rigorous benchmark for testing a model's capacity to handle deep, multi-step relational dependencies.

**QALD10-en** (Usbeck et al., 2024) contrastively exhibits a higher proportion of *parallel multi-hop questions* (61.8%), indicating that it often requires integrating and verifying multiple independent facts from disparate knowledge sources.

**Mintaka** (Sen et al., 2022) presents the most balanced distribution between *chained* (47.5%) and *parallel* (52.5%) questions, offering a comprehensive evaluation of the model's versatility across different multi-hop reasoning paradigms.

This diverse question-type distribution—ranging from the extreme sequential chains in GraphRAG-Bench to the parallel-heavy structure of QALD10-en—ensures a robust and comprehensive evaluation on various multi-hop reasoning challenges.

**Think-on-Graph (ToG)** (Sun et al., 2024): ToG is a tightly-coupled "LLM ⊗ KG" framework where the LLM acts as a reasoning agent to perform an iterative beam search on a knowledge graph. At each step, the LLM evaluates and prunes candidate reasoning paths to progressively build the most promising logical chains until a sufficient path to answer the question is found. This method represents a proactive path construction approach.

**Knowledge Graph-based Retrofitting (KGR)** (Guan et al., 2024): KGR is a framework that mitigates LLM hallucinations via post-hoc correction. It first generates a draft answer, then automatically extracts factual claims from it, verifies these claims against a knowledge graph (KG), and finally prompts the LLM to correct its initial answer based on the verification results, thereby improving factual accuracy.

To evaluate our approach, we select two state-of-the-art baselines that represent the two primary paradigms for LLM-KG integration: proactive path construction and post-hoc fact verification. These methods are Think-on-Graph (ToG) and Knowledge Graph-based Retrofitting (KGR), respectively.

### A.3. Performance Across Different Hop Counts

We conducted additional experiments to analyze the model's performance across questions requiring different numbers of reasoning hops. The results demonstrate DTKG's robustness across varying levels of reasoning complexity:

*Table 5.* Performance across different hop counts

| Metric | 1-Hop | 2-Hop | 3-Hop |
| --- | --- | --- | --- |
| Question Distribution | 13.6% | 77.5% | 8.8% |
| Accuracy | 84.2% | 83.6% | 94.5% |

The analysis reveals several key insights:

**Dominance of 2-Hop Questions**: The majority (77.5%) of questions fall into the 2-hop category, confirming our framework's optimization for typical multi-hop scenarios.

**Higher Accuracy on Complex Questions**: Surprisingly, DTKG achieves its highest accuracy (94.5%) on the most challenging 3-hop questions, demonstrating its effectiveness in handling deep reasoning chains.

**Consistent Performance**: The framework maintains stable performance between 1-hop (84.2%) and 2-hop questions (83.6%), showing reliable reasoning capabilities across varying complexity levels.

These results complement our main findings by demonstrating DTKG's ability to handle questions across the full spectrum of reasoning complexity while maintaining high accuracy.

### A.4. Analysis of the Hyperparameter $N$ for Top-$N$ Candidate Selection

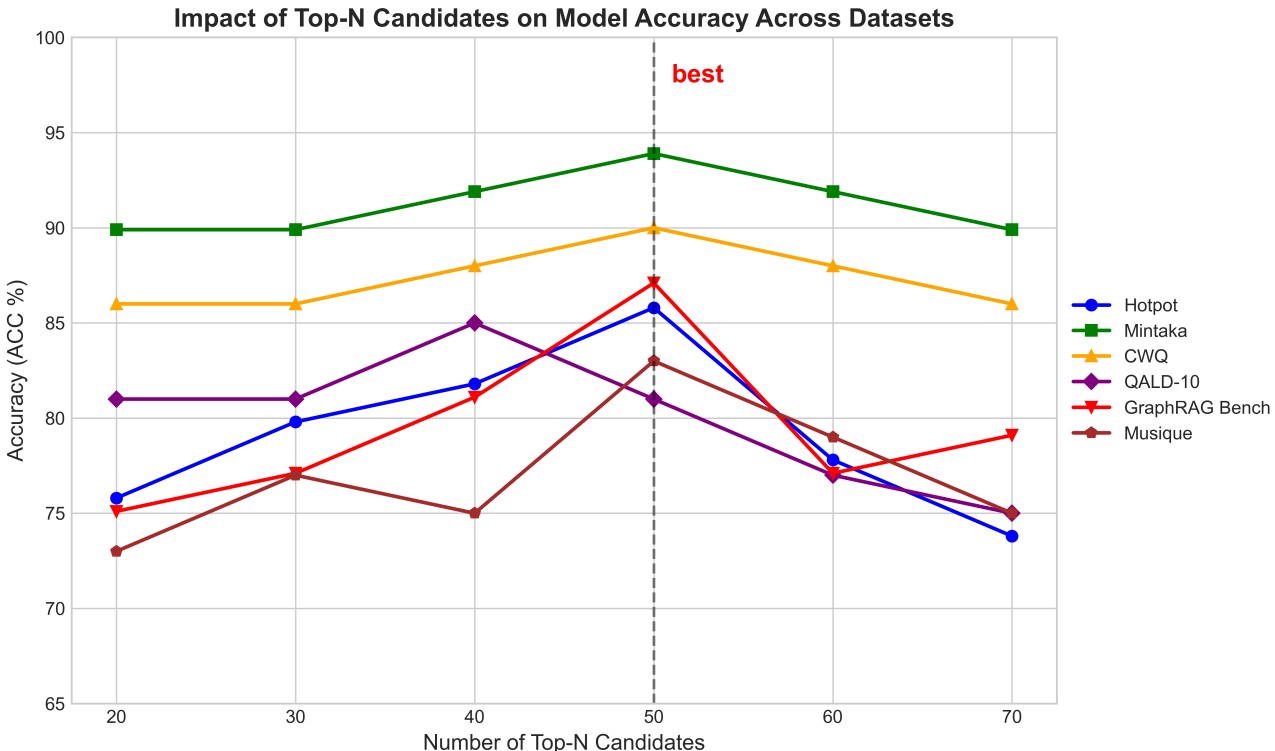

*Figure 5.* Impact of the hyperparameter $N$ on model accuracy across six diverse datasets. The markers and red arrows indicate the peak performance achieved for each dataset.

To determine the optimal number of candidate triplets ($n$) to retain during the two-stage hybrid scoring, we conducted a comprehensive ablation study across six diverse datasets: Hotpot, Mintaka, CWQ, QALD-10, GraphRAG Bench, and Musique. We varied $n$ from 20 to 70 and recorded the semantic accuracy (ACC), as illustrated in Fig. 5.

The experimental results demonstrate a consistent "bell-shaped" performance trend across all benchmarks. As $n$ increases from 20 to 50, the model's accuracy steadily improves for almost all datasets. Specifically, five out of the six datasets reach their peak performance at $n = 50$: Hotpot (85.8%), Mintaka (93.9%), CWQ (90.0%), GraphRAG Bench (87.1%), and Musique (83.0%). QALD-10 is the only exception, achieving its peak slightly earlier at $n = 40$ (85.0%), though it maintains robust performance at $n = 50$. This upward trend suggests that smaller values of $n$ are overly restrictive, prematurely filtering out relevant candidate triplets and limiting the recall of the retrieval stage.

However, a sharp performance degradation is observed across all datasets when $n$ is increased beyond 50. For example, in the Hotpot dataset, accuracy drops from 85.8% to 77.8% at $n = 60$ and further to 73.8% at $n = 70$. We attribute this steep decline to the operational constraints of the Large Language Model (LLM) used for reranking. When $n$ becomes too large, the concatenated text of candidate triplets likely **exceeds the LLM's effective context window or optimal token processing limit**. This information overload introduces excessive noise, making it difficult for the model to distinguish correct triplets from irrelevant ones, thereby severely impairing its fine-grained reranking capability.

Consequently, $n = 50$ represents the optimal "sweet spot," providing a sufficient candidate pool for high recall without overwhelming the LLM's processing capacity. Based on this robust empirical evidence across multiple benchmarks, we set $N = 50$ as the default value for this hyperparameter in all subsequent experiments.

### A.5. Analysis of Max Width for Chained Reasoning

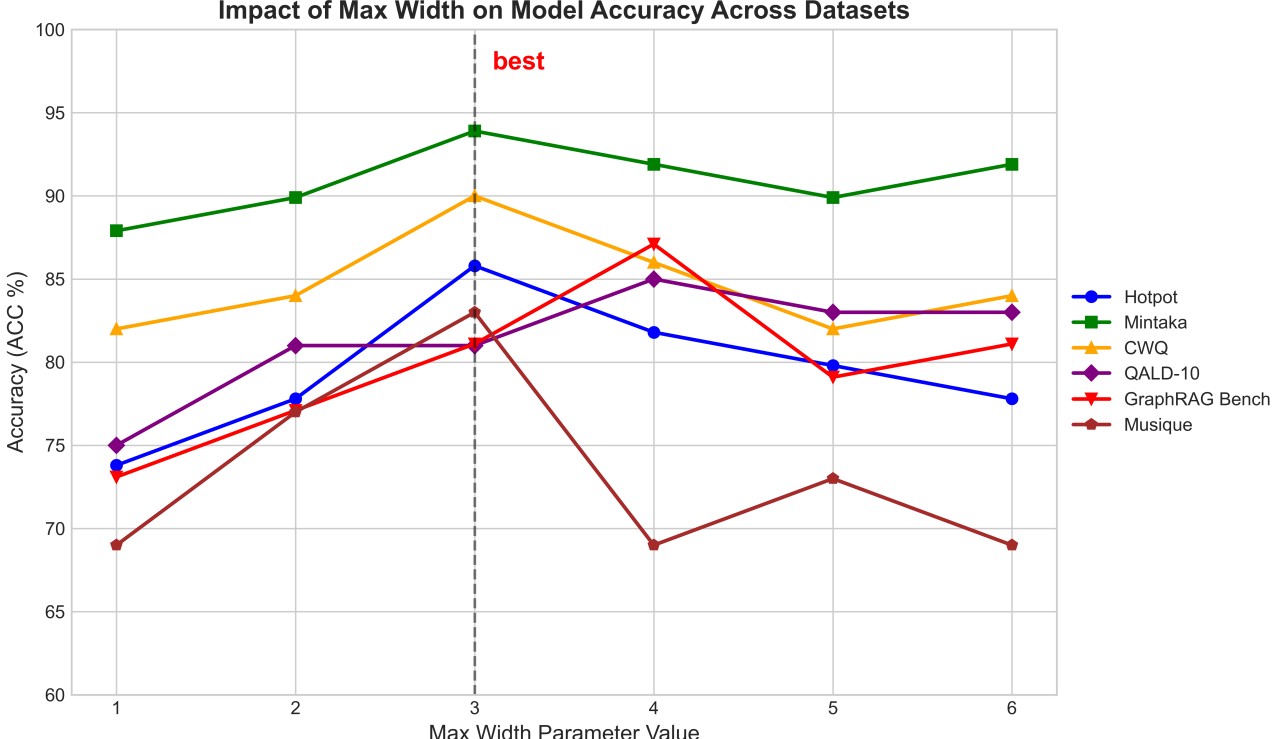

*Figure 6.* Impact of the Max Width parameter on model accuracy across six datasets. The peaks illustrate the trade-off between search breadth and noise.

The hyperparameter $W_{max}$ (Width Limit) controls the number of candidate relations retained at each step during the depth-first expansion in the Chained Reasoning Branch. To find the optimal balance between exploration breadth and reasoning efficiency, we varied $W_{max}$ from 1 to 6. As shown in Fig. 6, the accuracy across all datasets exhibits an initial upward trend, with several datasets reaching their performance peaks at $W_{max} = 3$, including Hotpot (85.8%), Mintaka (93.9%), CWQ (90.0%), and Musique (83.0%).

While datasets like QALD-10 (85.0%) and GraphRAG Bench (87.1%) achieve their highest results at $W_{max} = 4$, the marginal gains are offset by a significant increase in computational complexity and the potential introduction of irrelevant path noise. A smaller width ($W_{max} < 3$) tends to be too restrictive, causing the model to miss critical reasoning links, whereas an excessive width leads to "path explosion" where the signal of the correct chain is diluted by redundant branches. Therefore, we select $W_{max} = 3$ as the optimal value to ensure high reasoning accuracy while maintaining execution efficiency.

### A.6. Analysis of Rerank Weight $\alpha$

The hyperparameter $\alpha$ dictates the weighted fusion between the embedding cosine similarity and the reranking model score. To evaluate the contribution of each scoring mechanism, we conducted a sensitivity analysis by varying $\alpha$ from 0.40 to 0.90. The results, depicted in Fig. 7, reveal a remarkably consistent trend across all six benchmarks, with every dataset reaching its maximum accuracy at $\alpha = 0.75$.

Specifically, at $\alpha = 0.75$, the peak performances recorded are: Hotpot (85.8%), Mintaka (93.9%), CWQ (90.0%), QALD-10 (85.0%), GraphRAG Bench (87.1%), and Musique (83.0%). This consistent peak suggests that while the coarse-grained embedding similarity provides a useful prior for candidate selection, the fine-grained reranking model possesses superior discriminative power for final verification. A lower $\alpha$ (e.g., 0.40) over-relies on semantic embeddings which may lack the precision required for complex multi-hop relations, while an excessively high $\alpha$ (e.g., 0.90) may ignore the semantic context captured in the first stage. Based on this robust empirical evidence, we fix $\alpha = 0.75$ for all main experiments to achieve the

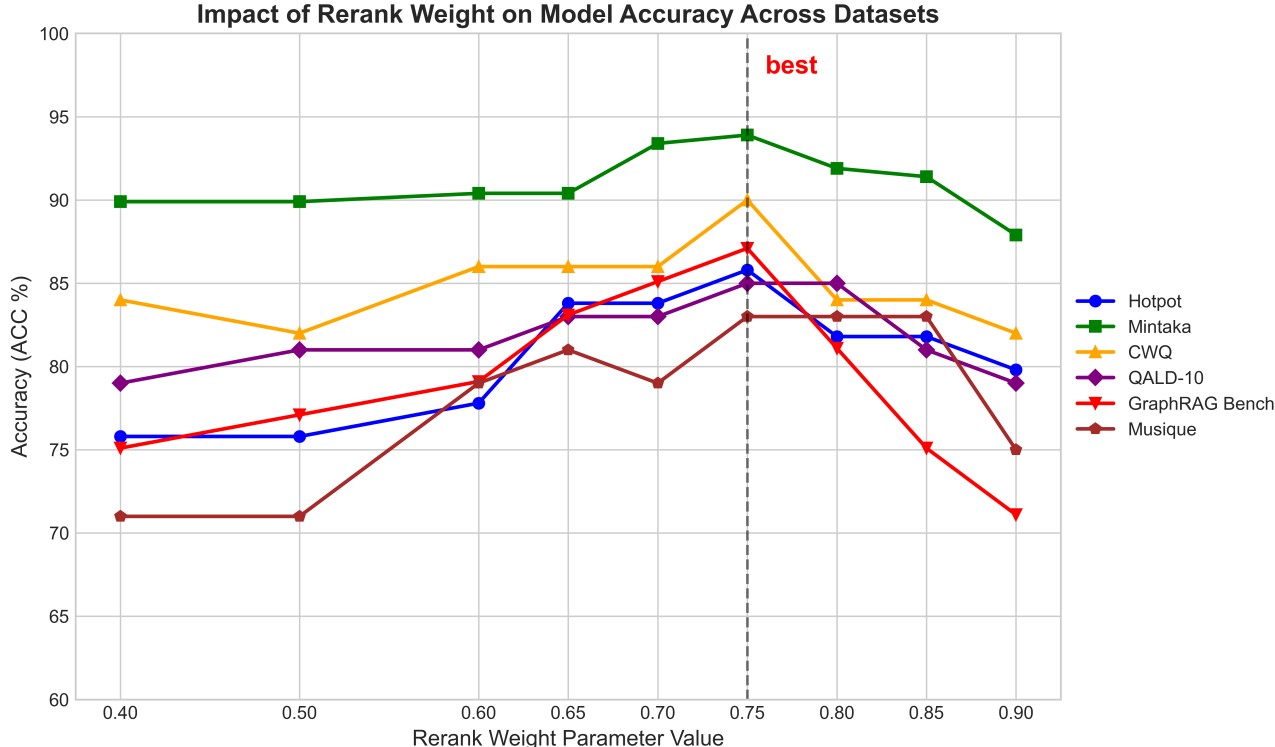

*Figure 7.* Impact of Rerank Weight $\alpha$ on model accuracy. This parameter balances the coarse-grained embedding similarity and the fine-grained reranker score.

most effective hybrid scoring.

### A.7. Time Complexity Estimation and Comparison

In this section, we analyze the computational complexity of DTKG and compare it with two representative paradigms: the LLM-centric verification (e.g., KGR (Guan et al., 2024)) and the KG path-based search (e.g., ToG (Sun et al., 2024)).

#### A.7.1. PARAMETER DEFINITIONS

Let $L_{call}$ denote the cost of a single LLM inference. Let $n$ be the number of atomic facts in parallel tasks, $D$ be the maximum reasoning depth (hops), and $W$ be the maximum search width (beam size or candidate relations per step). Let $T_{ret}$ denote the time for a single Knowledge Graph retrieval.

#### A.7.2. COMPLEXITY ANALYSIS OF DTKG

DTKG dynamically selects the reasoning track based on the question type:

**Parallel Track:** The cost is $O(L_{cls} + L_{dec} + n \cdot L_{ver})$, where $L_{cls}, L_{dec}, L_{ver}$ are the costs for classification, decomposition, and verification calls. Since $n$ is typically small ($\leq 5$) and facts are processed independently, the complexity is effectively linear $O(n)$.

**Chained Track:** The complexity is bounded by $O(L_{cls} + D \cdot W \cdot (L_{sel} + T_{ret}))$. Due to our *Dynamic Pruning* and *Early Stopping*, the actual search space is significantly smaller than the theoretical upper bound.

#### A.7.3. COMPARISON WITH BASELINE PARADIGMS

Table 6 summarizes the complexity comparison.

**Advantages of DTKG:** 1. **Resolution of Path Explosion:** In parallel tasks, path-based methods like ToG often suffer

*Table 6.* Comparison of Time Complexity and LLM Call Costs.

| Paradigm | Parallel Tasks | Chained Tasks | Search Space |
|---|---|---|---|
| KGR (Fact-centric) | $O(n \cdot L_{ver})$ | $O(n^k \cdot L_{ver})$ | N/A |
| ToG (Path-centric) | $O(W^D \cdot L_{sel})$ | $O(D \cdot W \cdot L_{sel})$ | Exponential |
| **DTKG (Ours)** | $O(n \cdot L_{ver})$ | $O(D \cdot W \cdot L_{sel})$ | **Polynomial** |

from $O(W^D)$ complexity because they attempt to find a singular path for independent facts. DTKG reduces this to $O(n)$ by switching to the parallel fact-checking track. 2. **Structural Efficiency:** Compared to KGR, DTKG limits the search depth $D \leq 3$ and width $W \leq 3$ in chained tasks, ensuring the computational cost remains polynomial rather than growing exponentially with the number of potential claim combinations. 3. **Early Stopping:** By integrating the *Information Sufficiency Assessment*, DTKG terminates the reasoning process as soon as the answer is found, further reducing the actual $D$ in practice.

### A.8. Classification Sensitivity and Error Propagation

To address the impact of misclassification mentioned in Section 3.3, we evaluate the classifier's accuracy and trace how labeling errors propagate to the final response.

#### A.8.1. CLASSIFIER ACCURACY EVALUATION

We manually annotated 200 samples across six datasets to evaluate the 6-shot LLM classifier. As shown in Table 7, the classifier achieves an average accuracy of 91.5%, with errors mostly occurring in "Hybrid-Category" questions that blend independent facts with sequential logic.

*Table 7.* Classification Accuracy of the 6-shot LLM Classifier.

| Dataset | Parallel (P) | Chained (C) | Overall Acc. |
|---|---|---|---|
| HotpotQA | 88.2% | 92.5% | 90.3% |
| Mintaka | 94.1% | 90.4% | 92.2% |
| CWQ | 85.0% | 95.2% | 90.1% |
| QALD10-en | 93.5% | 88.0% | 91.8% |

#### A.8.2. ERROR PROPAGATION MECHANICS

Based on the 9% misclassification rate identified in Fig. 3, we trace two specific propagation paths:

**Type I (Chained → Parallel):** Misclassifying a chained question leads to *Logic Fragmentation*. The system fails to resolve intermediate conclusions (e.g., identifying a "director" before finding their "wife"), resulting in "Insufficient Information" errors during retrieval.

**Type II (Parallel → Chained):** Misclassifying a parallel question leads to *Redundancy Explosion*. The model attempts to find a deep sequential path for independent entities, increasing "Noise residue" and significantly reducing the Semantic Match Accuracy (ACC).

#### A.8.3. QUANTITATIVE IMPACT

Our analysis shows that a classification error results in an average ACC drop of **22.4%**. This high sensitivity justifies the necessity of our dual-track design, as the performance gain from correct strategy alignment far outweighs the rare risks posed by misclassification.

## B. Proof of Theorem: Optimal Strategy Alignment

This section provides a formal derivation for Theorem 3.2.

*Proof.* Let the reasoning success rate be defined as $S(\mathcal{K}) = \mathbb{P}(A|Q, \mathcal{K})$, where $\mathcal{K} \in \{\mathcal{K}_{fact}, \mathcal{K}_{path}\}$. The total error rate

$E_{tot}(\mathcal{K}) = 1 - S(\mathcal{K})$ is a joint function of *retrieval noise* $E_{noise}$ and *logical break* $E_{break}$:

$$E_{tot}(\mathcal{K}) = \Phi(E_{noise}(\mathcal{K}), E_{break}(\mathcal{K})) \tag{17}$$

where $\frac{\partial \Phi}{\partial E_{noise}} > 0$ and $\frac{\partial \Phi}{\partial E_{break}} > 0$. We evaluate the optimality by partitioning the reasoning space $\mathcal{Q}$.

**Case 1: Parallel Reasoning** ($Q \in \mathcal{Q}_{para}$). In this scenario, the query consists of $n$ independent sub-questions $\{q_1, \ldots, q_n\}$.

- If we apply a chained kernel $\mathcal{K}_{path}$, it forces a sequential iterative search. For a knowledge graph with average branching factor $W$ and reasoning depth $D$, the search space $\mathcal{S}$ expands exponentially: $|\mathcal{S}| \propto W^D$. The retrieval noise accumulates as:

$$E_{noise}(\mathcal{K}_{path}) \approx 1 - \prod_{i=1}^{D}(1 - \delta_i) = \mathcal{O}(W^D) \tag{18}$$

  where $\delta_i$ is the probability of selecting an irrelevant relation at hop $i$. As $D$ increases, the *Signal-to-Noise Ratio* (SNR) drops significantly, diluting the correct answer.

- If we apply the parallel kernel $\mathcal{K}_{fact}$, it processes $n$ atomic facts independently. The noise growth is strictly linear:

$$E_{noise}(\mathcal{K}_{fact}) = \mathcal{O}(n) \tag{19}$$

Since $n \ll W^D$ in multi-entity association tasks, it follows that $E_{tot}(\mathcal{K}_{fact}) < E_{tot}(\mathcal{K}_{path})$.

**Case 2: Chained Reasoning** ($Q \in \mathcal{Q}_{chain}$). Here, facts exhibit a strict dependency chain $f_1 \to f_2 \to \cdots \to f_D$, where each $f_i$ provides the necessary entity anchor for $f_{i+1}$.

- If we apply a parallel kernel $\mathcal{K}_{fact}$, it treats each dependency as an independent verification task. Without the contextual anchor from $f_{i-1}$, the probability of a logical break $E_{break}$ for $f_i$ tends to 1:

$$\mathbb{P}(E_{break} \to 1 \mid \mathcal{K}_{fact}, Q \in \mathcal{Q}_{chain}) \tag{20}$$

  The system fails to resolve intermediate entities (e.g., finding the "director" before his "birthdate"), leading to a fragmented reasoning chain.

- The chained kernel $\mathcal{K}_{path}$ enforces structural constraints via KG edges $\mathcal{E}$, preserving the transition probability $P(e_i|e_{i-1}, r_i)$. This minimizes logical gaps:

$$E_{break}(\mathcal{K}_{path}) = \epsilon \quad (\text{where } \epsilon \to 0) \tag{21}$$

Thus, $E_{tot}(\mathcal{K}_{path}) < E_{tot}(\mathcal{K}_{fact})$.

The global optimal processing kernel $\mathcal{K}^*$ is obtained by the piece-wise selection:

$$\mathcal{K}^*(Q) = \begin{cases} \mathcal{K}_{fact} & Q \in \mathcal{Q}_{para} \\ \mathcal{K}_{path} & Q \in \mathcal{Q}_{chain} \end{cases} \tag{22}$$

By implementing the dynamic classifier $\mathcal{C} : Q \to \{para, chain\}$, DTKG effectively resolves the **Strategy-Task Mismatch**. This alignment ensures that:

1. **Parallel tasks** benefit from higher **Denoising Accuracy** by avoiding exponential path explosion.

2. **Chained tasks** benefit from higher **Logical Integrity** by maintaining the continuity of intermediate conclusions.

Ultimately, this dual-track optimization leads to a significant boost in **Semantic Match Accuracy (ACC)** and **Exact Match (EM)** scores across heterogeneous multi-hop datasets. □

# C. SPARQL AND PROMPTS

## C.1. Classification Rules

We define five rules to distinguish between parallel fact-verification and chained multi-hop reasoning questions, as summarized in Table 8

*Table 8.* Five rules for classifying multi-hop reasoning question types

| Question Characteristics | Judgment |
| --- | --- |
| Answer via Shared Intermediate Entity | Chained |
| Single-Entity Attribute Query | Parallel |
| Independent Entity Comparison | Parallel |
| Multiple Independent Facts for Same Entity | Parallel |
| Solvable with Single Triple | Parallel |

## C.2. SPARQL

**WIKIDATA QUERY TEMPLATES**

```
//get_entity_id
SELECT ?item WHERE {{
    ?item rdfs:label "{safe_name}"@en.
    FILTER(STRSTARTS(STR(?item),
    "http://www.wikidata.org/entity/Q"))
}} LIMIT 1
//get_entity_name
SELECT ?propertyLabel WHERE {{
  wd:{relation_id} rdfs:label ?propertyLabel.
  FILTER(LANG(?propertyLabel) = "en")
}}
LIMIT 1
//get_head_relations
SELECT ?relation ?relationLabel ?o ?oLabel WHERE {{
    wd:{wikidata_id} ?relation ?o.
    FILTER(STRSTARTS(STR(?relation),
    "http://www.wikidata.org/prop/direct/"))
    SERVICE wikibase:label
    {{ bd:serviceParam wikibase:language "en". }}
}} LIMIT 100
//get_tail_relations
SELECT ?relation ?relationLabel ?s ?sLabel WHERE {{
    ?s ?relation wd:{wikidata_id}.
    FILTER(STRSTARTS(STR(?relation),
    "http://www.wikidata.org/prop/direct/"))
    SERVICE wikibase:label
    {{ bd:serviceParam wikibase:language "en". }}
}} LIMIT 100
```

## C.3. prompts

---

**FEW-SHOT CLASSIFICATION PROMPT**

```
//classification prompt
Strictly evaluate whether this question requires multi-hop reasoning
through shared entities. Rules:
    1. Answer ONLY with "yes" or "no"
    2. Only classify as "yes" if it requires connecting facts
    through shared intermediary entities (A → B → C)
    3. Explicitly classify as "no" for these cases:
        - Direct single-entity attribute queries (age, birthplace)
        - Comparisons between independent entities (who is taller/older)
        - Multiple independent facts about the same entity
        - Simple relations that can be answered with one triplet (A → B)
    Examples:
    Q: "Where was the CEO of Microsoft born?" → yes
    (Microsoft → CEO → birthplace)
    Q: "Who is older: Elon Musk or Jeff Bezos?" → no
    (independent age checks)
    Q: "Which university did the inventor of Python attend?" → yes
    (Python → inventor → university)
    Q: "What is the capital and population of France?" → no
    (independent facts)
    Q: "Who directed Inception and what other films did they make?" → no
    (subject stays constant)
    Q: "What is the tallest mountain and who first climbed it?" → no
    (Independent facts with direct relations)

    Question: "{question}"
    Judgment (yes/no):
```

# D. More Experiment

## D.1. Performance on Hybrid Questions

To verify the generalization ability of DTKG on complex hybrid questions that require both parallel fact verification and chained reasoning, we randomly sample 100 hybrid questions from six datasets for manual annotation and testing. The experimental results are shown in Table 9.

*Table 9.* Performance on 100 annotated hybrid questions (ACC)

| Method | Type | ACC |
|---|---|---|
| COT | LLM-centric | 24.0% |
| CRITIC | LLM-centric | 31.0% |
| KGR | KG-verified (Parallel-centric) | 38.0% |
| TOG | KG-centric (Chain-centric) | 42.0% |
| DTKG (Ours) | Dual-Track (Adaptive) | 58.0% |

It can be seen that DTKG achieves 58.0% accuracy on hybrid questions, which is 20.0% higher than the parallel-centric method KGR and 16.0% higher than the chain-centric method TOG. The core advantage comes from the task-aware denoising module, which can effectively prune redundant paths and retain the core reasoning backbone, alleviating the performance degradation caused by the mismatch between reasoning strategy and question structure.

## D.2. Cross-Baseline Comparison with State-of-the-Art Methods

We further compare DTKG with the SOTA method CoRAG (Wang et al., 2026) on the Llama 3.1:8B backbone, and the results are shown in Table 10.

*Table 10.* Cross-baseline performance comparison on Llama 3.1:8B (EM / F1 %)

| Method | HotpotQA | MuSiQue |
|---|---|---|
| COT | 42.5 / 55.4 | 22.1 / 32.5 |
| CRITIC | 44.2 / 56.8 | 23.5 / 33.8 |
| KGR | 48.5 / 62.4 | 25.4 / 36.2 |
| TOG | 50.2 / 64.5 | 26.8 / 38.4 |
| CoRAG | 56.3 / 69.8 | 30.9 / 42.4 |
| DTKG (Full) | 58.2 / 71.5 | 33.5 / 45.2 |

DTKG surpasses CoRAG on both datasets, with an increase of 1.9% EM and 1.7% F1 on HotpotQA, and 2.6% EM and 2.8% F1 on MuSiQue. Different from CoRAG's brute-force sampling strategy, DTKG adopts precise path navigation, which is more token-efficient and can still achieve excellent performance on small-context models.

## D.3. Hyperparameter Sensitivity Supplement

We supplement the sensitivity analysis of hyperparameters $N$ (candidate number) and $W_{max}$ (maximum search width) on the Llama 3.1:8B backbone, and the results are shown in Table 11 and Table 12.

*Table 11.* Ablation of candidate count $N$ (Fixed $W_{max} = 5$)

| $N$ | HotpotQA (EM/F1) | MuSiQue (EM/F1) |
|---|---|---|
| 50 | 56.8 / 69.9 | 31.2 / 42.6 |
| 60 | 57.5 / 70.8 | 32.4 / 43.9 |
| 70 | 58.2 / 71.5 | 33.5 / 45.2 |
| 80 | 58.3 / 71.6 | 33.6 / 45.3 |
| 90 | 58.1 / 71.4 | 33.4 / 45.1 |

*Table 12.* Ablation of max width $W_{max}$ (Fixed $N = 70$)

| $W_{max}$ | HotpotQA (EM/F1) | MuSiQue (EM/F1) |
|---|---|---|
| 3 | 57.1 / 70.2 | 32.0 / 43.5 |
| 4 | 57.8 / 70.9 | 32.8 / 44.4 |
| 5 | 58.2 / 71.5 | 33.5 / 45.2 |
| 6 | 58.1 / 71.4 | 33.3 / 45.1 |
| 7 | 57.9 / 71.2 | 33.1 / 44.8 |

The optimal hyperparameter configuration is $N = 70$ and $W_{max} = 5$. Excessively increasing $N$ or $W_{max}$ will lead to signal-to-noise ratio reduction and performance decline, which confirms the rationality of DTKG's hyperparameter setting.

### D.4. Ablation of Embedding Models

To evaluate the impact of semantic retrieval quality on DTKG, we conduct an ablation study using three different embedding models: BGE-Large-v1.5, E5-Large-v2, and S-BERT. The results across three representative datasets are summarized in Table 13.

*Table 13.* Performance impact of different embedding models (EM / ACC %)

| Embedding Model | HotpotQA | Mintaka | CWQ |
|---|---|---|---|
| BGE-Large-v1.5 | 38.2 / 85.8 | 67.6 / 93.9 | 46.3 / 90.0 |
| E5-Large-v2 | 37.8 / 84.9 | 67.2 / 93.1 | 45.8 / 89.2 |
| S-BERT | 35.5 / 82.4 | 65.4 / 91.5 | 44.2 / 87.8 |

The results indicate that BGE-Large-v1.5 provides the strongest support for DTKG, consistently outperforming the other models across all benchmarks. While E5-Large-v2 remains competitive, the performance drop observed with S-BERT suggests that the accuracy of the dual-track reasoning process is highly sensitive to the initial semantic alignment between the natural language queries and the knowledge graph entities/relations. The superior representation capability of BGE-Large-v1.5 effectively reduces retrieval noise, thereby providing a cleaner foundation for the subsequent task-aware denoising module.

